# Effect of Replacing Corn Meal with Winged Bean Tuber (*Psophocarpus tetragonolobus*) Pellet on Gas Production, Ruminal Fermentation, and Degradability Using In Vitro Gas Technique

**DOI:** 10.3390/ani14030356

**Published:** 2024-01-23

**Authors:** Pachara Srichompoo, Chaichana Suriyapha, Chanon Suntara, Sompong Chankaew, Teppratan Rakvong, Anusorn Cherdthong

**Affiliations:** 1Tropical Feed Resources Research and Development Center (TROFREC), Department of Animal Science, Faculty of Agriculture, Khon Kaen University, Khon Kaen 40002, Thailand; pachara.s@kkumail.com (P.S.); chaichana_s@kkumail.com (C.S.); chansun@kku.ac.th (C.S.); 2Department of Agronomy, Faculty of Agriculture, Khon Kaen University, Khon Kaen 40002, Thailand; somchan@kku.ac.th (S.C.); teppratan_r@kkumail.com (T.R.)

**Keywords:** energy feed source, alternative feedstuffs, pelleting process, ruminal fermentation

## Abstract

**Simple Summary:**

The increasing expenses of animal feed, which are affected by the prices of protein and carbohydrates, emphasize the need for alternate sources of protein. Thailand is now doing research on tropical legumes, with a special focus on winged beans. The tubers of winged beans, which are rich in carbohydrates and protein, provide a viable alternative to animal feed. The process of pelletizing winged bean tubers has many benefits, including enhanced durability and prolonged preservation. This makes it a feasible substitute for corn meal in animal feed. The research assesses the replacement of corn meal with winged bean tuber pellets in ruminant diets, examining its impact on ruminal fermentation, gas production, and in vitro degradability.

**Abstract:**

The objective of this study is to evaluate the effects of replacing corn meal in ruminant diets with winged bean (*Psophocarpus tetragonolobus*) tubers (WBT) on ruminal fermentation, gas production parameters, and in vitro degradability. The study employed a completely random design (CRD) in its execution. The experimental design employed was a completely randomized design (CRD), featuring eleven levels of corn meal substitution with winged bean tubers pellet (WBTP) at 0%, 10%, 20%, 30%, 40%, 50%, 60%, 70%, 80%, 90%, and 100%. The levels were grouped into four categories of replacement: control (0% in the diet), low levels (10%, 20%, and 30% in the diet), medium levels (40%, 50%, 60%, and 70% in the diet), and high levels (80%, 90%, and 100% in the diet). The experimental results indicated that substituting corn meal with WBTP at moderate and high levels in the diet could improve the performance of the fermentation process by increasing the gas production rate constant from the insoluble fraction (*p* < 0.01). The IVDMD exhibited a higher degree of in vitro degradation after 12 h (h), with the mean value being higher in the high group compared to the medium until the high group (*p* < 0.05). At the 4 h mark, the groups that substituted corn meal with WBTP exhibited a decrease in pH value (*p* < 0.05) in comparison to the control group. The substitution of corn meal with WBTP resulted in the lowest protozoal count after 8 h in the median group (*p* < 0.05). A significant difference in the effect of WBTP on total volatile fatty acid (TVFA) concentration was observed at 8 h after incubation (*p* < 0.05). The medium and high levels of WBTP replacement resulted in the lowest TVFA concentration at 8 h (*p* < 0.05). The mean proportion of acetic acid (C2) linearly declined and was lowest when a high level of WBTP replaced cornmeal (*p* < 0.05). The concentration of propionic acid (C3) at 8 h after incubation and average values were linearly significantly different when various levels of WBTP were utilized. Replacing corn meal with WBTP at a high level showed the highest concentration of C3. Moreover, substituting medium and high concentrations of WBTP for corn meal resulted in a significant reduction in both the C2:C3 ratio at 8 h and the mean value (*p* < 0.05). In conclusion, WBTP exhibits a nutritional composition that is advantageous and may be an energetic substitute for corn meal.

## 1. Introduction

The ongoing dispute between Russia and Ukraine, the two largest exporters of animal feed in the world, has had an additional detrimental impact on ensuring the protection of the global fodder supply [1]. As a result of its heavy reliance on imported feed ingredients, Thailand is being severely impacted by this economic downturn. The Thai Feed Mill Association (TFMs) states that between 2021 and 2022, the cost of animal feed in Thailand increased by 30.5%, 56.6%, and 52.9%, respectively, for corn, wheat, and barley. Nevertheless, feed price experts have observed that while the Russia–Ukraine conflict has resulted in immediate price increases, it offers a prospect for a sustained transition toward feed resources in the long run [2]. In developing nations, in particular, the demand for animal-derived feed has increased due to the ever-expanding global population [3]. Present-day food-feed materials contend on an international scale as both animal feed and human sustenance, frequently comprising identical constituents. Elevated feed prices have been fueled by protein and carbohydrate sources, including corn, cassava, and legumes. Consequently, this has sparked renewed interest in exploring alternative nitrogen sources for livestock feed. Incorporating novel tuberous plant species or alternative plant species has the potential to enhance local feed diversity and alleviate feed limitations in specific areas [4].

Tropical legumes, such as the winged bean (*Psophocarpus tetragonolobus*; WB), are utilized in ruminant diets only infrequently [5]. This tropical legume vegetable possesses significant nutritional value and is cultivated in various tropical regions, Thailand being among them [6]. The edible components of the WB, including its tuberous tubers, seeds, leaves, and blossoms, are present in every province of Thailand. In addition, the WB tubers (WBT) are high in carbohydrates (25–30%) and contain up to 20% protein [5,7], which are suitable as an alternative animal feed source. Recently, Suntara et al. [8] revealed WBT could be utilized as an alternative energy feed source in place of cassava chips in the diet of ruminants if farmers were to recognize its benefits; this could ultimately contribute to the expansion of animal feed production on a large scale. Moreover, Unnawong et al. [9] demonstrate that, as rumen-resistant starch, the steam-treated WBT exhibited a decrease in both the soluble fraction (a) and degradation rate.

The pelletizing process increases the density of feed and decreases the occurrence of stratification loss, both of which are advantageous for the nutritional value and potential of the feed. Furthermore, the pelleting procedure may result in the loss of a portion of the CP content as a consequence of elevated temperatures. Nevertheless, the pelletizing process could potentially offer several benefits, including improved feeding stability, prolonged storage duration, nutritional integrity preservation, and access to granules that possess supplementary nutritional attributes [10,11].

We hypothesized that, similar to the particle feed produced by WBT, it might be suitable for application as animal feed and serve as a viable substitute energy source for feedstuffs. In vitro gas production was utilized to examine the consequences of substituting pellet feed from WBT (WBTP) for corn meal in this study.

## 2. Materials and Methods

To ensure animal welfare, the Animal Ethics Committee of Khon Kean University authorized the experimental cattle used in this study (record number IACUC-KKU-98/66).

### 2.1. Location, Winged Bean Tubers, and Winged Bean Tuber Pellet

The research was carried out at Khon Kaen University (KKU), Faculty of Agriculture, Department of Animal Science, Khon Kaen, Thailand. The WBT was cultivated and harvested at the Faculty of Agriculture, Department of Agronomy, KKU. After being chopped to a length of 3–5 cm, the obtained WBT was sun-dried for 72 h to achieve a dry matter (DM) of at least 90%. Dried WBT was powered through a 1 mm screen using a Cyclotech Mill (Tecator, Hoganas, Sweden) for use in the pellet. The WBT pellet was made using 100 kg of ground WBT and 25 L of water, and it was pelleted using a Ryuzoukun mini with a 50.8 mm thick die and a 4.8 mm diameter hole (Kakiuchi Co., Ltd., Kochi, Japan). Finally, the WBT pellets (WBTP) were sun-dried for around 3 days to ensure an acceptable moisture content of 8–10% [11].

### 2.2. Dietary Treatments and Experimental Design

The experimental design employed was a completely randomized design (CRD), featuring eleven levels of corn meal substitution with WBTP at 0%, 10%, 20%, 30%, 40%, 50%, 60%, 70%, 80%, 90%, and 100%. The levels were grouped into four categories of replacement: control (0% in the diet), low levels (10%, 20%, and 30% in the diet), medium levels (40%, 50%, 60%, and 70% in the diet), and high levels (80%, 90%, and 100% in the diet). Following standard procedures, the desiccated dietary samples were all passed through a 1 mm sieve (Cyclotech Mill, Tecator, Hoganas, Sweden) at 60 °C in order to ascertain the dry matter (DM-viz. 34.01) and organic matter (OM-viz. 942.05) content [12]. The crude protein (CP) content was analyzed using a nitrogen analyzer (Leco FP828 Nitrogen Analyzer, LECO Corporation, Saint Joseph, MI, USA). The fiber was analyzed in the following manner: neutral-detergent fiber (NDF), acid-detergent fiber (ADF), and acid-detergent lignin (ADL) by following the method of Van Soest et al. [13]. The gross energy (GE) contents of all experimental diets were determined in an autocalculating bomb calorimeter (SHIMADZU CA-4PJ, SHIMADZU Corporation, Kyoto, Japan). The chemical composition of experimental diets is presented in Table 1.

### 2.3. Animal Donors and Ruminal Inoculum Preparation

Fluid from the rumen was obtained from 3 male Thai native beef cattle, each of which weighed 280 ± 15.0 kg at birth. The animals were fed a concentrate diet at 1% of body weight daily at 6:30 a.m. and 4:30 p.m. The concentrate diet comprised 14.0% crude protein (CP) and 75.5% total digestible nutrients (TDN), including cassava chip, corn meal, rice bran, soybean meal, palm kernel meal, molasses, urea, salt, minerals, and vitamins. Additionally, they had ad libitum access to rice straw at all times. The cattle were housed in secluded enclosures and were supplied with mineral blocks and potable water on a regular basis. For twenty-one days, the cattle were fed the regimens. Prior to morning feeding, a suction device was employed to collect 2000 mL of ruminal fluid from each cattle. The rumen fluid from the three animals was mixed and provided as a representative sample from the rumen fluid donors. Following the filtration process using four layers of cheesecloth, the resultant sample was transferred into an Erlenmeyer vial. Proceeded to the laboratory using preheated thermos vessels. The WBTP and corn meal were combined with the other components of the concentrate diet, as shown in Table 1. The in vitro substrate test consisted of 0.5 g of dry matter, which had a ratio of 40% roughage to 60% concentrated diet. Rice straw was employed as a source of roughage. The roughage and concentrate were ground and sieved through a 1 mm mesh (Cyclotech Mill, Tecator, Hoganas, Sweden) before being added to the bottle test. The experimental feed was weighed into bottles of 50 mL each. After sealing the canisters with aluminum and rubber closures, oxygen (O_2_) was removed by purging them with carbon dioxide (CO_2_). An artificial saliva formulation was prepared in accordance with the methodology outlined by Menke et al. [14]. This involved the combination of distilled water, microminerals, macrominerals, and a buffer solution. Subsequently, for the ruminal inoculum mixed medium, artificial saliva and rumen fluid were extensively mixed at a 1:2 ratio (artificial saliva: rumen fluid) while continuous CO_2_ purging occurred [14]. The 50 mL vials containing 0.5 g of dietary treatment were weighed. Using a 1.5-inch, 18-gauge needle, 40 mL of the ruminal inoculum mixed medium was transferred into each bottle. The bottles were then incubated at 39 °C for a duration of 96 h.

### 2.4. In Vitro Gas Production and Fermentation Characteristics

The experimental units were as follows: The kinetics of gas were assessed with three replicates per treatment (3 bottles/treatment × 11 treatments = 33 bottles). From each measured value, the five blank bottles were subtracted to determine the net gas production. A total of 66 bottles (3 bottles/treatment × 11 treatments × 2 sampling times (4 and 8 h incubation)) were separately prepared for pH, NH_3_-N, volatile fatty acids (VFAs), and protozoal count analysis. Another set of 66 bottles (3 bottles/treatment × 11 treatments × 2 sampling times (12 and 24 h incubation)) was prepared for digestibility analysis. Three groups were established for the experimental bottles; the initial group was designated for gas kinetics and production. The quantity of gas generated was ascertained through the monitoring and documentation of the volume of gas produced subsequent to incubation using a 20 mL glass hypodermic syringe affixed to the incubation container at a distance of 1.5 inches. Gas emission was quantified using a glass syringe in the identical incubation vessel at the following time intervals: 0, 0.5, 1, 2, 3, 4, 5, 6, 7, 8, 9, 10, 11, 12, 18, 24, 48, 72, and 96 h subsequent to incubation. The gas production quantity was computed by employing the models of Ørskov and McDonald [15], which were used for curve fitting and analysis of the kinetics of gas as follows:y = a + b (1 − e^(−ct)^)
where a = soluble fraction from gas production, b = insoluble fraction from gas production, c = rate of gas production constant for the insoluble fraction (b), t = incubate time, (|a| + b) = the potential extent of gas production, and y = gas produced at the time ‘t’.

Ruminal parameter measurements, including pH, were conducted on the second group at 4 and 8 h after incubation using a Hanna Instruments (HI) 8424 microcomputer manufactured by Hanna Instruments, Inc. in Kallang, Singapore. The incubated ruminal liquor was subsequently partitioned into two portions. The initial portion (20 mL) was reconstituted in 5 mL of 1 M H_2_SO_4_ and preserved at −20 °C for analysis of ammonia nitrogen (NH_3_-N) using a spectrophotometer (UV/Vis Spectrometer, PG Instruments Ltd., London, UK) in accordance with the procedure described by Fawcett and Scott [16]. The ruminal in vitro fluid samples were analyzed for volatile fatty acids (VFA), including acetic acid (C2), propionic acid (C3), and butyric acid (C4). A gas chromatography apparatus was utilized, a Newis GC-2030 manufactured by Shimadzu Corporation in Kyoto, Japan, in conjunction with a capillary column (DB-Wax column, 30 m length, 0.25 mm diameter, 0.25 m film; Agilent, Santa Clara, CA, USA), as described by Porter and Murray [17]. In order to directly enumerate protozoa, the second portion (1 mL) was withdrawn into 9 mL of a 10% formalin solution [18]. The third group is for measuring in vitro degradability. Following incubation for 12 and 24 h, the in vitro dry matter degradability (IVDMD) and in vitro organic matter degradability (IVOMD) were assessed utilizing the protocols outlined by Tilley and Terry [19].

### 2.5. Statistical Analysis

A completely random design (CRD) arrangement was utilized to statistically evaluate the experiment’s data in accordance with CRD using the Proc. GLM procedure of SAS software version 9.4 [20]. System for Statistical Analysis (SAS). The analysis of all data was conducted using the subsequent equation:Y_ij_ = μ + τ_i_ + ε_ij_
where Y_ij_ is the dependent variable, μ is the overall mean of the experiment, τ is the variable due to the group or treatment, and ε is the variable of random error. Duncan’s New Multiple Range Test was utilized to compare treatment methods in numerous ways. Orthogonal polynomials for diet responses were determined by linear, quadratic, and cubic effects. Variations in the mean values that were less than or equal to *p* < 0.05 were regarded as statistically significant.

## 3. Results

### 3.1. Kinetics and Cumulative Production of Gas

The impact of substituting winged bean tuber pellets (WBTP) for corn on gas kinetics and cumulative gas at 96 h postincubation is illustrated in Table 2. There were no observable impacts on the cumulative gas production, gas production from the immediately soluble fraction (a), gas production from the insoluble fraction (b), or the overall gas potential represented by the sum of gas production from both fractions (|a| + b) (*p* > 0.05). However, the gas production rate constant for the insoluble fraction (c) was linearly increased in the high-level group (*p* < 0.01), with a range of 0.06–0.09 mL/h.

### 3.2. In Vitro Degradability

Table 3 illustrates the impact of substituting corn with WBTP on in vitro degradability. There was a greater in vitro degradability of IVDMD at 12 h and the mean value in the high group and the medium until the high-level group, respectively (*p* < 0.01). Meanwhile, the IVDMD at 24 h was not changed by dietary treatments (*p* > 0.05). In contrast, compared to the other dietary interventions, IVOMD demonstrated the lowest value in the control group (0% corn meal replacement with WBTP) (*p* < 0.05).

### 3.3. In Vitro Ruminal pH, Ammonia Nitrogen Concentration, and Protozoal Number

Table 4 presents the impact of substituting corn with WBTP on pH levels, NH_3_-N concentration, and protozoa populations. At 4 h, the groups that substituted corn with WBTP (at low, medium, and high levels) exhibited a decrease in pH of fermentation fluid at sampling time (*p* < 0.05) compared to the control group (which received no WBTP replacement). In contrast, both the pH value at 8 h and the mean value remained constant. (*p* > 0.05). Furthermore, no significant difference was observed in the number of protozoa at 4 h after incubation (*p* > 0.05). The protozoal count at 8 h was shown as the lowest number (4.18 × 10^6^ cell/mL) when replacing corn with WBTP in the medium-level group, while the protozoal count at the mean value was shown as the highest number (6.55 × 10^6^ cell/mL) in the control group (0% of replacing corn with WBTP) (*p* < 0.05). However, the research did not observe a significant change in the concentration of NH_3_-N (*p* > 0.05).

### 3.4. In Vitro Volatile Fatty Acids

The impact of substituting WBTP for corn on the properties of volatile fatty acids (VFA) is illustrated in Table 5. A significant difference in the effect of WBTP on total VFA (TVFA) concentration was observed at 8 h after incubation (*p* < 0.05). The medium and high levels of WBTP replacement resulted in the lowest TVFA concentration at 8 h (*p* < 0.05), while the average TVFA concentration did not change (*p* > 0.05) when treated with the control group. The mean proportion of C2 linearly declined and was lowest when a high level of WBTP replaced cornmeal (*p* < 0.05). The concentration of C3 at 8 h after incubation and average values were linearly significantly different when various levels of WBTP were utilized. Replacing corn meal with WBTP at a high level showed the highest concentration of C3. Moreover, substituting medium and high concentrations of WBTP for corn meal resulted in a significant reduction in both the C2:C3 ratio at 8 h and the mean value (*p* < 0.05).

## 4. Discussions

### 4.1. Kinetics and Cumulative Production of Gas

The findings of this research indicate that the potential extent of gas production (|a| + b) and the similar cumulative gas production are all likely attributable to the high starch content of the WBTP. This starch content may render the WBTP similar to corn in terms of facilitating digestion and generating specific beneficial gases. In addition, the replacement group did not differ in gas generation from the insoluble fraction (b). If NDF and ADF levels are similar across different diet levels, it implies that the microbial population in the rumen has a consistent amount of fibrous material to ferment [21]. This stable substrate availability may result in relatively constant gas production from the fermentation of the insoluble fraction [22]. Zarski et al. [23] It has been documented that modifying the processes of starch production, including processes, pressure roasting, pelletizing, extrusion, particle size, fiber content, and others, can significantly affect the degradation of starch in compound feeds within the rumen. In the present inquiry, negative values for the instantaneously soluble fraction (a) signify a deviation from the expected exponential outcome of fermentation or a delay in the commencement of fermentation. The delayed time before microbial colonization occurs This is due to the fact that fermentation of the cell walls has not yet begun subsequent to the soluble fraction of the substrate being consumed [22,23]. The previous studies also reported negative values for various substrates when mathematical models were used to match the kinetics of gas emission [23,24]. As a result, the mathematical model that represents the fermentation of the soluble fraction and adequately characterizes gas production kinetics utilizes the absolute value of a (|a|) [22,25]. It is understood that it is possible to use the absolute value to define the ideal fermentation of the soluble fraction. Similarly, Suntara et al. [24] reported that the WBT could replace cassava chips up to 100% in diet without negative impact on in vitro gas cumulative and kinetics of gas. Furthermore, the higher gas production rate constant from the insoluble fraction (c) at medium or high levels may be attributed to the processing of particles into pellets during the experiment, which has the potential to enhance the available surface area for microbial attack, leading to improved substrate fermentation [25,26]. Pelletized starch can become gelatinized at certain temperatures and for certain amounts of time during processing, which can increase the amount of fermentation [27]. Similarly, Solanas et al. [25] demonstrated a connection between the level of starch gelatinization and the rise in an in vitro fractional gas production rate.

### 4.2. In Vitro Degradability

In this study, a higher IVDMD was observed at 12 h, along with an increased mean value when WBTP was used to replace corn at a high level in the diet. This could be due to the fact that the substitution of corn with WBTP positively influenced the early-stage digestibility of dry matter. The enhanced IVDMD at 12 h indicates that ruminants fed diets containing WBTP may experience enhanced nutrient utilization and overall feed efficiency during the initial stages of digestion [27]. The fermentation rate of carbohydrates is influenced by factors such as monosaccharide and bond composition, molecular size, sugar arrangement at the molecular level, and physical morphology [28]. In addition, the gelatinization of pellet processing has the potential to enhance the available surface area for microbial attack, leading to improved degradability [25,28]. However, the IVDMD at 24 h showed no significant changes among the dietary treatments. This finding suggests that the benefits of replacing corn with WBTP may be more pronounced in the earlier stages of digestion, with the effect diminishing over time. Thus, it is essential to consider the practical implications of this result, as it may influence feeding recommendations and ration formulation for ruminant animals over extended periods. Furthermore, the present study revealed a higher IVOMD value in the group where corn was replaced with WBTP as compared to the control group. Furthermore, the physical properties of addition feed pellets, which are produced by densifying constituent mixtures into particles, must be evaluated to ensure that their design, management, and transportation systems are optimized [29]. This finding is valuable as it suggests that WBTP may contribute positively to the overall nutritional quality of the diet, promoting better utilization of organic matter by ruminants.

### 4.3. In Vitro Ruminal pH, Ammonia Nitrogen Concentration, and Protozoal Number

The ruminal pH ranged from 6.70 to 6.78 in the current investigation, which falls within the established range of rumen ecology (pH 6.20 to 7.00) [30]. The decrease in pH at 4 h indicates that the dietary changes had an immediate impact on rumen pH. However, the lack of significant changes in pH at 8 h may indicate that the rumen microbial population adapted to the dietary shift over time. Agarwal et al. [31] stated that rumen microbial activity is dependent on specific pH conditions that vary with rumen conditions and that normal rumen pH ranges from 6.0 to 7.0 for normal rumen metabolism. Throughout fermentation, the population of active microorganisms may be influenced by the acidity level in the rumen [32].

The NH_3_-N concentrations observed in this investigation fell within the expected range of 15–30 mg/dL. This finding holds promise for enhancing ruminal feed digestibility and facilitating microbial protein synthesis [33]. However, the NH_3_-N concentration remained unchanged. This phenomenon may be attributed to the insignificant impact of the modified starch on the protein degradation rate [34]. Suntara et al. [24] revealed that the levels of WBT replacing cassava chips in the diet had no impact on in vitro NH_3_-N concentration.

The protozoal count at the 8 h mark exhibited a decline as the concentration of WBTP increased at medium and high levels in comparison to the control group. Pelletized feeds, commonly subject to processing and compression, often yield a feed with a more consistent and finely textured structure [11,26]. The physical characteristics and processing methods of pellets may impact the accessibility of starch and sugar, which serve as primary nutrient sources for protozoal growth [18]. Rumen protozoa are recognized for their preference for fermentable carbohydrate sources, and a transition to more processed feeds could potentially diminish the availability of these materials, resulting in a reduction in protozoal populations [18]. Additionally, it was hypothesized that the presence of tannins, saponins, and other phenolic compounds in WBT might contribute to the reduction of protozoal numbers [35,36]. However, due to limitations in resources and the specific objectives of our study, the analysis did not ascertain the concentrations of these compounds. *P. tetragonolobus* is commonly consumed in various forms, such as raw or cooked leaves, flowers, tubers, and pods, and has been reported to exhibit anti-inflammatory, antioxidant, and antinociceptive activities [35]. Prior investigations have suggested that saponins and tannins possess the potential to suppress protozoa populations, facilitating the development of bacteria in the rumen [37,38]. Similarly, Ampapon et al. [39] illustrated that supplementing the diet with a tropical plant (*Amaranthus cruentus*, L.) rich in phytonutrients, including tannins and saponins, resulted in reduced protozoal populations and in vitro methane production.

### 4.4. In Vitro Volatile Fatty Acids

Total volatile fatty acid (TVFA) concentration at 8 h peaked at the control group and then declined at the high replacement level. The processing of pelletized WBTP can influence the reduction in in vitro total VFA concentration through several factors related to the physical and chemical changes that occur during the pelletization process [39]. The compression and heat involved in pelleting may cause starch gelatinization or alterations in nutrient structure, impacting the accessibility of nutrients for VFA synthesis. Additionally, WBTP feeds may have a different carbohydrate composition compared to corn or other nonpelleted feeds. The type and proportion of carbohydrates in the diet can influence the profile of VFAs produced during fermentation. If pellet feeds contain carbohydrates that are more rapidly fermented, it may lead to a decrease in total VFA concentration. Suntara et al. [8] reported that replacing cassava chips with WBT could potentially reduce TVFA levels by 20%.

In addition, when considering VFA profiles, WBT contains a high carbohydrate content, which may contribute to C3 production [5,7]. The present experiment demonstrates that replacing cornmeal with WBTP at a high level resulted in the highest C3 production. Thus, an increase in C3 concentration leads to a reduced proportion of C2. Propionate is a gluconeogenic VFA, meaning it can be converted to glucose in the liver. Glucose serves as a vital energy source for the host animal, particularly in functions such as growth, lactation, and reproduction [7]. An increase in C3 ate production contributes to greater glucose availability. A significant component of rumen fermentation, the proportion C2:C3 in ruminant diets affects nutrient utilization and the microbial community [9]. Increased supply efficiency is frequently associated with a reduced ratio of C2:C3. Enhanced production of C3, which is a VFA with a higher energy density than C2, may facilitate improved energy utilization from diet [9,11]. This study indicates that the proportion of C2 and the C2:C3 ratio decreases, while the C3 and C4 proportions increase with the rising level of WBTP in the diet. It is possible that the high temperature during pellet or steam-flaking processing disrupted the disulfide bond of proteins and the embedded structure of starch. This alteration could render starch a more accessible surface for ruminal microbial degradation enzymes. However, when the C2:C3 ratio is drastically reduced, the availability of precursors for milk lipid synthesis may be altered, which could have an effect on the composition and yield of the milk [24,26]. Previous studies reported that steam-flaking cereal grains led to an increase in C3 concentration and a decrease in the ratio of C2 to C3 [39,40]. However, the potential cause may be attributed to the presence of tannin and saponin compounds in WBT, which may have stimulated the growth of *F. succinogenes*, a microbe recognized for succinate production in the rumen [41]. The increased production of C3 can be linked to the succinate–propionate pathway, a primary route for C3 generation in the rumen [42]. Phesatcha et al. [43] showed that incorporating mangosteen peel, which is rich in tannin and saponin, into the diet of dairy cows stimulated the growth of *F. succinogenes* and increased the proportion of C3.

## 5. Conclusions

In conclusion, WBTP possesses a valuable nutritional composition and could serve as an alternative feed source. The replacement of corn with WBTP at a high level has no negative impact on in vitro gas production or the kinetics of gas and enhances degradability, as well as increasing the proportion of C3 and C4 while decreasing the C2 proportion and the C2:C3 ratio. However, an increase in the level of WBTP in the diet resulted in lower TVFA production at 8 h and the mean value. Thus, additional in vivo investigation is required to ascertain the true intestinal accessibility of WBTP during animal experimentation.

## Figures and Tables

**Table 1 animals-14-00356-t001:** Ingredients and chemical composition of corn, winged bean tubers pellet, rice straw, and experimental concentrate diets.

Ingredients (%DM)	Replacement Levels											Corn Meal	WBTP ^1^	RS ^2^
	T1 (0%)	T2 (10%)	T3 (20%)	T4 (30%)	T5 (40%)	T6 (50%)	T7 (60%)	T8 (70%)	T9 (80%)	T10 (90%)	T11 (100%)			
Cassava chips ^3^	40.0	40.0	40.0	40.0	40.0	40.0	40.0	40.0	40.0	40.0	40.0	-	-	-
Corn meal	10.0	9.0	8.0	7.0	6.0	5.0	4.0	3.0	2.0	1.0	0.0	-	-	-
Rice bran	14.0	14.0	14.0	14.0	14.0	14.0	14.0	14.0	14.0	14.0	14.0	-	-	-
WBTP	0.0	1.0	2.0	3.0	4.0	5.0	6.0	7.0	8.0	9.0	10.0	-	-	-
Soybean meal	14.0	14.0	14.0	14.0	14.0	14.0	14.0	14.0	14.0	14.0	14.0	-	-	-
Palm kernel meal	16.0	16.0	16.0	16.1	16.2	16.2	16.2	16.3	16.3	16.4	16.4	-	-	-
Molasses	3.0	3.0	3.0	3.0	3.0	3.0	3.0	3.0	3.0	3.0	3.0	-	-	-
Urea	1.0	1.0	0.9	0.9	0.8	0.8	0.8	0.7	0.7	0.6	0.6	-	-	-
Salt	1.0	1.0	1.0	1.0	1.0	1.0	1.0	1.0	1.0	1.0	1.0	-	-	-
Minerals and vitamins ^4^	1.0	1.0	1.0	1.0	1.0	1.0	1.0	1.0	1.0	1.0	1.0	-	-	-
Chemical composition														
Dry matter (%)	88.2	87.3	86.5	85.6	84.8	84.8	83.1	82.3	81.6	81.6	80	87.5	42.1	94.2
Organic matter (% DM)	95.1	95.1	95.2	95.2	95.2	95.2	95.3	95.3	95.3	95.4	95.4	96.2	94.5	86.2
Crude protein (% DM)	14.1	14.1	14.1	14.2	14.2	14.2	14.2	14.2	14.2	14.2	14.2	7.9	15.7	2.4
Neutral detergent fiber (% DM)	20.7	21.5	22.1	23.1	23.7	24.2	24.8	25.1	25.3	25.9	26.3	22.1	24.7	64.3
Acid detergent fiber (% DM)	10.4	10.4	10.5	10.5	10.6	10.6	10.6	10.7	10.7	10.7	10.8	8.3	9.4	40.2
Gross energy (Mcal/kg DM)	4.6	4.5	4.5	4.5	4.5	4.5	4.4	4.4	4.4	4.4	4.4	3.4	1.2	3.8

^1^ WBTP = winged bean tubers pellet, ^2^ RS = rice straw, ^3^ Cassava chip contained: 90.17% day matter, 2.04% crude protein, 0.77% ether extract, 27.5% neutral-detergent fiber, 4.01% acid-detergent fiber, 66.74% non-fiber carbohydrate, ^4^ Minerals and vitamins (per kg): Vitamin A = 10,000,000 IU, Vitamin E = 70,000 IU, Vitamin D = 1,600,000 IU, Fe = 50 g, Zn = 40 g, Mn = 40 g, Co = 0.1 g, Cu = 10 g, Se = 0.1 g, I: 0.5 g.

**Table 2 animals-14-00356-t002:** Effect of replacing corn with winged bean tuber pellets on gas kinetics and cumulative gas at 96 h after incubation.

Treatment	Gas Kinetics				Cumulative Gas (mL)
	a	b	c	|a| + b	
Control	−8.82	98.78	0.06 ^e^	107.60	90.50
Low ^1^	−9.07	115.42	0.06 ^e^	124.49	106.25
Medium ^2^	−8.91	99.42	0.07 ^de^	108.33	90.41
High ^3^	−9.09	85.90	0.09 ^d^	94.99	73.78
SEM	0.21	2.40	0.00	2.30	3.09
Contrast					
Linear	0.78	0.06	0.01	0.06	0.07
Quadratic	0.98	0.09	0.56	0.08	0.13
Cubic	0.71	0.13	0.90	0.12	0.24

^1^ Low = average effect of replacing corn with winged bean tuber pellets at levels of 10, 20 and 30% in diet, ^2^ Medium = average effect of replacing corn with winged bean tuber pellets at levels of 40, 50, 60 and 70% in diet, ^3^ High = average effect of replacing corn with winged bean tuber pellets at levels of 80, 90, and 100% in diet, ^d,e^ Means with different letters in a column are significantly different at *p* < 0.05, SEM = standard error of the means, a = the gas production from the immediately soluble fraction, b = the gas production from the insoluble fraction, c = the gas production rate constant from the insoluble fraction, (|a| + b) = the gas potential extent of gas production.

**Table 3 animals-14-00356-t003:** Effect of replacing corn with winged bean tuber pellets on in vitro degradability.

Treatment	IVDMD, %DM			IVOMD, %DM		
	12 h	24 h	Mean	12 h	24 h	Mean
Control	51.04 ^c^	63.85	57.45 ^bc^	87.70 ^c^	88.84 ^c^	88.27 ^c^
Low ^1^	50.99 ^c^	62.41	56.70 ^c^	89.92 ^a^	91.68 ^a^	90.80 ^a^
Medium ^2^	54.32 ^b^	61.78	58.05 ^ab^	89.70 ^a^	90.48 ^b^	90.09 ^b^
High ^3^	57.54 ^a^	60.76	59.15 ^a^	88.93 ^b^	91.56 ^a^	90.25 ^ab^
SEM	0.20	0.30	0.21	0.10	0.06	0.08
Contrast						
Linear	<0.01	0.06	0.03	0.06	<0.01	<0.01
Quadratic	0.10	0.81	0.27	<0.01	<0.01	<0.01
Cubic	0.16	0.13	0.09	0.03	<0.01	<0.01

^1^ Low = average effect of replacing corn with winged bean tuber pellets at levels of 10, 20 and 30% in diet, ^2^ Medium = average effect of replacing corn with winged bean tuber pellets at levels of 40, 50, 60 and 70% in diet, ^3^ High = average effect of replacing corn with winged bean tuber pellets at levels of 80, 90, and 100% in diet, ^a,b,c^ Means with different letters in a column are significantly different at *p* < 0.05, SEM = standard error of the means, IVDMD = In vitro dry matter degradability, IVOMD = In vitro organic matter degradability.

**Table 4 animals-14-00356-t004:** Effect of replacing corn with winged bean tuber pellets on ruminal pH, ammonia nitrogen concentration, and protozoal number of fermentation fluid at sampling time.

Treatment	Ruminal pH			Ammonia Nitrogen, mg/dL			Protozoal Number, ×10^6^ cell/mL		
	4 h	8 h	Mean	4 h	8 h	Mean	4 h	8 h	Mean
Control	6.78 ^a^	6.72	6.75	19.03	29.17	24.10	7.50	5.60 ^a^	6.55 ^a^
Low ^1^	6.76 ^b^	6.70	6.73	17.26	24.84	21.05	6.50	5.54 ^a^	6.02 ^ab^
Medium ^2^	6.76 ^b^	6.71	6.74	17.61	28.84	23.23	7.13	4.18 ^c^	5.66 ^b^
High ^3^	6.76 ^b^	6.70	6.73	17.63	25.16	21.40	6.34	4.84 ^b^	5.58 ^b^
SEM	0.001	0.004	0.002	0.63	0.87	0.75	0.18	0.02	0.09
Contrast									
Linear	0.04	0.34	0.24	0.60	0.43	0.50	0.21	<0.01	0.03
Quadratic	0.03	0.58	0.18	0.55	0.89	0.86	0.95	<0.01	0.18
Cubic	0.40	0.33	0.85	0.58	0.12	0.23	0.14	<0.01	0.70

^1^ Low = average effect of replacing corn with winged bean tuber pellets at levels of 10, 20 and 30% in diet, ^2^ Medium = average effect of replacing corn with winged bean tuber pellets at levels of 40, 50, 60 and 70% in diet, ^3^ High = average effect of replacing corn with winged bean tuber pellets at levels of 80, 90, and 100% in diet, ^a,b,c^ Means with different letters in a column are significantly different at *p* < 0.05, SEM = standard error of the means.

**Table 5 animals-14-00356-t005:** Effect of replacing corn with winged bean tuber pellets on volatile fatty acid (VFA) characteristics.

Treatment	Total VFA, mM			Acetic Acid (A), mol/100 mol			Propionic Acid (P), mol/100 mol			Butyric Acid, mol/100 mol			A:P Ratio		
	4 h	8 h	Mean	4 h	8 h	Mean	4 h	8 h	Mean	4 h	8 h	Mean	4 h	8 h	Mean
Control	67.51	84.36 ^a^	75.93	64.81	68.64 ^a^	66.73 ^a^	27.25	23.51 ^b^	25.38 ^b^	7.95	7.85 ^b^	7.91 ^b^	2.38	2.92 ^a^	2.66 ^a^
Low ^1^	70.80	79.08 ^ab^	74.94	65.31	68.52 ^a^	66.92 ^a^	26.91	23.27 ^b^	25.14 ^b^	7.79	8.11 ^b^	7.95 ^b^	2.32	2.66 ^b^	2.54 ^ab^
Medium ^2^	65.85	77.67 ^b^	71.76	65.23	67.37 ^a^	66.30 ^a^	26.75	25.47 ^b^	25.47 ^b^	8.03	8.49 ^b^	8.21 ^ab^	2.44	2.46 ^c^	2.39 ^b^
High ^3^	69.83	75.25 ^b^	72.54	64.92	62.59 ^b^	63.76 ^b^	26.72	28.02 ^a^	27.37 ^a^	8.37	9.39 ^a^	8.88 ^a^	2.47	2.28 ^c^	2.38 ^b^
SEM	0.75	0.74	0.59	0.35	0.87	0.22	0.29	0.32	0.08	0.10	0.89	0.08	0.04	0.02	0.03
Contrast															
Linear	0.87	<0.05	0.12	0.99	<0.05	<0.01	0.55	<0.05	<0.05	0.21	<0.05	<0.05	0.76	<0.01	<0.05
Quadratic	0.57	0.36	0.36	0.62	0.12	<0.01	0.75	0.10	0.06	0.45	0.31	0.32	0.55	0.20	0.34
Cubic	0.12	0.35	0.53	0.83	0.69	0.37	0.90	0.74	0.77	0.71	0.59	0.97	0.43	0.39	0.88

^1^ Low = average effect of replacing corn with winged bean tuber pellets at levels of 10, 20 and 30% in diet, ^2^ Medium = average effect of replacing corn with winged bean tuber pellets at levels of 40, 50, 60 and 70% in diet, ^3^ High = average effect of replacing corn with winged bean tuber pellets at levels of 80, 90, and 100% in diet, ^a,b,c^ Means with different letters in a column are significantly different at *p* < 0.05, A:P ratio = acetic acid to propionic acid ratio, SEM = standard error of the means.

## Data Availability

The data that support the findings of this study are available from the corresponding author upon reasonable request.

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
