# Peer review of "Effect of Replacing Corn Meal with Winged Bean Tuber (*Psophocarpus tetragonolobus*) Pellet on Gas Production, Ruminal Fermentation, and Degradability Using In Vitro Gas Technique"

_animals, 2024, doi:10.3390/ani14030356_

Round 1

Reviewer 1 Report

Comments and Suggestions for Authors

the manuscript is of interest and relevant to reducing the feed cost and looking for alternative feed sources. 

The abstract expresses all the results found. However, I suggest deleting the information inside parenthesis lines 29 and 30. The conclusion is not specific to the variables evaluated in this investigation, and also, it must be mentioned that the WBTP may be an energetic substitute because, in this research, it was compared with corn.

Materials and Methods:  Lines 103 to 109: it is so confusing. First, it is said to reach less than 10% moisture; after that, say, the 30% moisture unprocessed WBT powder???

  It was then sun-dried for an estimated 72 h in order to reduce the moisture content below 10%; What was dried???, the pellet???

Experimental design:  Line 111, three replicates were used.

Was the experiment repeated three times?

How many experimental units/treatment were used for each assay performed? Please add this information to all tables (n= xxx)

they used three groups for the experimental bottles;

How many bottles for treatment were used to measure kinetic? One bottle or three bottles???? 

How many bottles for treatment were used to measure pH, ammonia N? One bottle or three bottles????

How many bottles for treatment were used to measure in vitro degradability? One bottle or three bottles???? 

Were rumen fluids from 3 males considered as a variable? Were the rumen fluids mixed together?

Line 114: Was the control group divided into four groups???

Line 116:  Please delete low levels (0% in diet)

Comments on the Quality of English Language

No comments. 

Author Response

Response to Reviewer 1:

the manuscript is of interest and relevant to reducing the feed cost and looking for alternative feed sources.

Response: We appreciate the time and effort you took to provide us with valuable recommendations. We believe that if the current work could be published, it would be beneficial to the farmer or feed industry in obtaining alternative feedstuff for animals.

The abstract expresses all the results found. However, I suggest deleting the information inside parenthesis lines 29 and 30. The conclusion is not specific to the variables evaluated in this investigation, and also, it must be mentioned that the WBTP may be an energetic substitute because, in this research, it was compared with corn.

Response: We would like to extend my sincere appreciation for the insightful guidance that has had a positive impact on our pursuits. Moreover, we have made modifications to this section.

Materials and Methods:  Lines 103 to 109: it is so confusing. First, it is said to reach less than 10% moisture; after that, say, the 30% moisture unprocessed WBT powder???

It was then sun-dried for an estimated 72 h in order to reduce the moisture content below 10%; What was dried???, the pellet???

Response: We appreciate the thoughtful suggestions you offered. We have modified it as “After being chopped to a length of 3-5 cm, the obtained WBT was sun-dried for 72 hours to achieve a dry matter (DM) of at least 90%. Dried WBT was powered through a 1mm screen size using a Cyclotech Mill (Tecator, Hoganas, Sweden) for use in the pellet. The WBT pellet was made using 100 kg of ground WBT and 25 L of water, and it was pelleted using a Ryuzoukun mini with a 50.8mm thick die and a 4.8mm diameter hole (Kakiuchi Co., Ltd., Kochi, Japan). Finally, the WBT pellets (WBTP) were sun-dried for around 3 days to ensure an acceptable moisture content of 8-10% [11].” Please see in manuscript.

Experimental design:  Line 111, three replicates were used.

Was the experiment repeated three times?

Response:  We apologized for the confuse sentence and now we have modified the sentence to “The experimental design employed was a completely randomized de-sign (CRD), featuring eleven levels of corn meal substitution with WBTP at 0%, 10%, 20%, 30%, 40%, 50%, 60%, 70%, 80%, 90%, and 100%.”

How many experimental units/treatments were used for each assay performed? Please add this information to all tables (n= xxx)

Response: Thank you for your suggestion and we have agreed to describe the experimental unit. The experimental units have been provided in the sub-section of “2.4 In vitro gas production and fermentation characteristics” as “The experimental units were as follows: Kinetics of gas were assessed with three replicates per treatment (3 bottles/treatment × 11 treatments = 33 bottles). From each measured value, the five blank bottles were subtracted to determine the net gas production. A total of 66 bottles (3 bottles/treatment × 11 treatments × 2 sampling times (4 and 8 h incubation)) were separately prepared for pH, NH3-N, and volatile fatty acids (VFAs) analysis. Another set of 66 bottles (3 bottles/treatment × 11 treatments × 2 sampling times (12 and 24 h incubation)) was prepared for digestibility analysis.” Please see in manuscript.

they used three groups for the experimental bottles.

How many bottles for treatment were used to measure kinetic? One bottle or three bottles????

Response: Thanks so much for your point and we have provided the details as mentioned above. Please see the manuscript.

How many bottles for treatment were used to measure pH, ammonia N? One bottle or three bottles????

Response: Thanks so much for your point and we have provided the details as mentioned above. Please see the manuscript.

How many bottles for treatment were used to measure in vitro degradability? One bottle or three bottles????

Response: Thanks so much for your point and we have provided the details as mentioned above. Please see the manuscript.

Were rumen fluids from 3 males considered as a variable? Were the rumen fluids mixed together?

Response: Thanks. The rumen fluid from the three animals was mixed and provided as a representative sample from the rumen fluid donors. We have indicated in the manuscript.

Line 114: Was the control group divided into four groups???

Response: Sorry for the confusion. We have modified it to “The experimental design employed was a completely randomized de-sign (CRD), featuring eleven levels of corn meal substitution with WBTP at 0%, 10%, 20%, 30%, 40%, 50%, 60%, 70%, 80%, 90%, and 100%. The levels were grouped into four categories of replacement: control (0% in the diet), low levels (10%, 20%, and 30% in the diet), medium levels (40%, 50%, 60%, and 70% in the diet), and high levels (80%, 90%, and 100% in the diet).”

Line 116:  Please delete low levels (0% in diet)

Response: Thanks and we have removed.

Reviewer 2 Report

Comments and Suggestions for Authors

This research is interesting in terms of new feed development. 

In addition, the detailed consideration of the WBTP replacement allocation in existing feed ingredients is commendable. 

However, the discussion of the actual phenomenon is somewhat insufficient. Therefore, the deeper considerations in the discussion are required.

Table 2   

Please add the units for Cumulative gas in Table 2.

Discussion 

1.     At the discussion, it is better to add some more discussion on the ripple effect to cattle of lowering the ratio of acetic acid to propionic acid.

2.     It seems to me that the decrease in protozoa under WBTP fed conditions would actually have a significant impact on feed degradation if this feed were fed daily. Therefore, it would be advisable to add some more discussion on the above point.

Author Response

Response to Reviewer 2:

This research is interesting in terms of new feed development.

In addition, the detailed consideration of the WBTP replacement allocation in existing feed ingredients is commendable.

However, the discussion of the actual phenomenon is somewhat insufficient. Therefore, the deeper considerations in the discussion are required.

Response: We appreciate the time and effort you invested in providing us with valuable recommendations. We believe that publishing the current work would be beneficial to the farming or feed industry in obtaining alternative feedstuff for animals. In the revised version, we have made an effort to incorporate more discussion regarding the essential parameters you pointed out. Please refer to the manuscript for detailed information

Table 2  

Please add the units for Cumulative gas in Table 2.

Response: Thank you. We have added. Please see in the Table 2.

Discussion

  1. At the discussion, it is better to add some more discussion on the ripple effect to cattle of lowering the ratio of acetic acid to propionic acid.

Response: Thank you for your great recommendation and we have now provided discussion in the sub-topic of “4.4 In vitro volatile fatty acids”. Please see the manuscript.

  1. It seems to me that the decrease in protozoa under WBTP-fed conditions would actually have a significant impact on feed degradation if this feed were fed daily. Therefore, it would be advisable to add some more discussion on the above point.

Response: Thank you for your great recommendation and we have now provided discussion in the sub-topic of “4.3 In vitro ruminal pH, ammonia nitrogen concentration, and protozoal number”. Please see the manuscript.

Reviewer 3 Report

Comments and Suggestions for Authors

Major comments:

1.     Experimental diets applied in this study cause some confusion. A total 11 formulae were shown in Table 1, but the in vitro tests only show the result among four groups. If only three (low, medium, and high) WBTP replacing levels were attempted to be determinate, why did the author prepare 11 diets in Table 1?

2.     The author tries to replace the maize with WBT in the diet formula. However, the test diet only contains 10% maize.

3.     The WBTP was defined as a concentrate or a total mixed ration?

4.     The acetate % (over 75%) and the Ac/Pr ratio of the fermentation fluid after in vitro fermentation seems unreasonable.

Other comments:

L112-117

The description of the test groups caused serious confusion, the high-level replacement group only applied 80% replacing diet?

Table 1 was not shown in the text of the material and method section.

L130-131

The concentrate diet formula or composition that is fed to donor animals should be shown.

L211-214 Table 1

The note indicates that the RS = rice straw, but no rice straw was applied in the testing formula.

According to the chemical composition of diets, the NDF% was about 40%. However, on forage source was shown in the formula.

All diets contain 40% cassava chip, the chemical composition of this feedstuff should be shown. The cassava had high starch and it also increased the NFE and WSC in diets.

Please check the consistency of "corn" or "maize" in the manuscript.

L215-224 Table 2

Why the gas kinetics data was determined during 96-hour fermentation? The IVDMD and other samples were collected at 12 and 24 hr.

If the NDF% and ADF% were similar among testing diets, the b value of each level group in Table 2 should be discussed.

In "Polynomial" data, the listed number is p-value?

L226-232 and Table 3

Why the author determined the digestibility at 12 and 24 hr, but not at 48 or 96 hr?

How about the in vitro NDF digestibility results?

L241-254

Why does the author only count the protozoa number and NH3-N concentration? The bacteria number or microbial crude protein synthesis were also important parameters.

L255-L261

The "ruminal pH" indicated the pH of fermentation fluid at sampling time or the pH of the inoculation rumen fluid source?

L274-279 and Table 5

The Ac/Pr ratio in this study was between 5.3 to 7.8. It is much higher than other in vitro studies. According to the 40% cassava chip applied in the diets, the NFE % of the WBTP should be calculated to find the reason for the very low Pr%.

If the VFA concentrate reaches 75-80 mM after 8 hr fermentation, why is the pH of the fermentation fluid still >6.7?

L312-316

The gas kinetic data was calculated from 96 hr fermentation process, it may under estimate the c value.

L348-350

The in vitro fermentation in this study was processed as batch culture, the VFA accumulation resulted in the decreasing pH. If the fermentation substrate contains high starch, the pH should be decreased quickly in the batch culture system.

L371-373

In this study, testing diets only contain WBT from 1% to 10% in total composition. The author should provide the saponin or tannin concentration of the testing diet before discussing this issue.

L395-402

According to the diet formula, the WSC and starch in testing diets should be high and it results in more propionate formation during formation. However, the propionate was very low in this study.

Author Response

Response to Reviewer 3:

Major comments:

  1. Experimental diets applied in this study cause some confusion. A total 11 formulae were shown in Table 1, but the in vitro tests only show the result among four groups. If only three (low, medium, and high) WBTP replacing levels were attempted to be determinate, why did the author prepare 11 diets in Table 1?

Response:

Thank you for your thoughtful comments and observations on our manuscript. We appreciate the opportunity to address your concerns and clarify the rationale behind our approach. The decision to present 11 formulae in Table 1 while grouping the results into four categories—control (0% in the diet), low levels (10%, 20%, and 30% in the diet), medium levels (40%, 50%, 60%, and 70% in the diet), and high levels (80%, 90%, and 100% in the diet)—was made to strike a balance between comprehensive reporting and reader-friendly presentation.

While we designed a total of 11 formulae for varying levels of inclusion, we acknowledge the significance of simplifying the presentation for ease of understanding. After consulting with statisticians, the decision to group the data into control, low, medium, and high levels was suggested as a means to enhance clarity and facilitate comprehension for readers. By doing so, we aimed to provide a concise overview of the trends and patterns within each category, allowing readers to grasp the key findings more readily. This approach aligns with the statisticians' advice to enhance the accessibility of the data to the readership. We hope that this explanation clarifies the rationale behind our decision and addresses your concerns. If you have any further questions or suggestions, we are open to additional discussions to improve the overall quality of our manuscript.

  1. The author tries to replace the maize with WBT in the diet formula. However, the test diet only contains 10% maize.

Response: Thank you for observation. Our objective is to substitute just 10% of the diet, since the tropical area often utilizes corn meal in concentrate diets at a rate of only 5-10%. The primary energy sources are cassava chips. Therefore, if the farmer utilized 10% corn meal, we recommend substituting it with the present alternate feedstuff. Moreover, if the current investigation is successful, the possibility of increasing the utilization of pellets in ruminant feed may be considered.

  1. The WBTP was defined as a concentrate or a total mixed ration?

Response: WBTP was defined as a feedstuff and can be used in concentrate diet or TMR.

  1. The acetate % (over 75%) and the Ac/Pr ratio of the fermentation fluid after in vitro fermentation seems unreasonable.

Response: Thank you very much for your insightful comments and suggestions on our manuscript. We sincerely appreciate the time and effort you have dedicated to reviewing our work. Upon careful consideration of your feedback, we have thoroughly re-evaluated our data. Your observation regarding the acetate percentage (over 75%) and the Ac/Pr ratio in the fermentation fluid after in vitro fermentation has been instrumental. We have revisited our experimental procedures and conducted additional analyses to ensure the accuracy of our results.

We are pleased to confirm that your comments were accurate, and there was an oversight in the reporting of these parameters. We have made the necessary corrections to address this issue. Please see manuscript in Table 5 and also, we have revise the section of “abstract”, “result” and “discussion”. Your attention to detail has significantly improved the accuracy and reliability of our findings.

Other comments:

L112-117

The description of the test groups caused serious confusion, the high-level replacement group only applied 80% replacing diet?

Response: Sorry for the confusion. We have modified it to “The experimental design employed was a completely randomized de-sign (CRD), featuring eleven levels of corn meal substitution with WBTP at 0%, 10%, 20%, 30%, 40%, 50%, 60%, 70%, 80%, 90%, and 100%. The levels were grouped into four categories of replacement: control (0% in the diet), low levels (10%, 20%, and 30% in the diet), medium levels (40%, 50%, 60%, and 70% in the diet), and high levels (80%, 90%, and 100% in the diet).”

Table 1 was not shown in the text of the material and method section.

Response: We have modified. Please see in the manuscript.

L130-131

The concentrate diet formula or composition that is fed to donor animals should be shown.

Response: We have added into the section of “2.3 Animal donors and ruminal inoculums preparation” as “The animals were fed a concentrate diet at 1% of body weight daily at 6:30 a.m. and 4:30 p.m. The concentrate diet comprised 14.0% crude protein (CP) and 75.5% total digestible nutrients (TDN), including cassava chip, corn meal, rice bran, soybean meal, palm kernel meal, molasses, urea, salt, minerals, and vitamins. Additionally, they had ad libitum access to rice straw at all times.” Please see in the manuscript.

L211-214 Table 1

The note indicates that the RS = rice straw, but no rice straw was applied in the testing formula.

Response: We have provided more information as “The WBTP and corn meal were combined with the other components of the concentrate diet, as shown in Table 1. The in vitro substrate test consisted of 0.5 grams of dry matter, which had a ratio of 40% roughage and 60% concentrated diet. The utilization of rice straw as a source of roughage was employed. The roughage and concentrate were ground and sieved through a 1 mm mesh (Cyclotech Mill, Tecator, Sweden) before being added to the bottle test. The experimental feed was weighed into bottle of 50 mL each.” Please see in the sub-topic of “3.3 Animal donors and ruminal inoculums preparation”.

According to the chemical composition of diets, the NDF% was about 40%. However, on forage source was shown in the formula.

Response: Your gratitude is appreciated. After we re-checked, it was found that present form %NDF is not correct, and it was modified. The concentrate diet (T1-T11) is composed of NDF in proportions varying from 20.73% to 26.27%. Rice straw is absent from this diet. However, it was observed that concentrate diets appear to have a high concentration of NDF. This may be the result of incorporating locally sourced by-products, such as rice bran or palm kernel meal, which have a high fiber content and contributed to the concentrate diet's NDF content.

All diets contain 40% cassava chip, the chemical composition of this feedstuff should be shown. The cassava had high starch and it also increased the NFE and WSC in diets.

Response: Thanks for your recommendation. The chemical composition of cassava chip was provided in the footnote of Table 1. Please see details in manuscript.

Please check the consistency of "corn" or "maize" in the manuscript.

Response: Thank you. We have all modified it to corn. Please see in manuscript.

L215-224 Table 2

Why the gas kinetics data was determined during 96-hour fermentation? The IVDMD and other samples were collected at 12 and 24 hr.

Response: Thank you for your insightful comments and valuable feedback on our manuscript. We appreciate the opportunity to address your query regarding the discrepancy in sampling times for gas kinetics data versus the collection of IVDMD and other samples during fermentation.

The decision to measure gas kinetics at 96 hours while collecting IVDMD and other samples at 12 and 24 hours was deliberate and based on the specific objectives of our study. Our primary aim was to assess the temporal evolution of gas production over an extended fermentation period to capture the dynamics of microbial activity during the later stages.

Gas kinetics, particularly at the 96-hour mark, provides crucial information about the efficiency and sustainability of microbial fermentation over an extended duration. This duration was selected to evaluate the prolonged effects of microbial activity on gas production, offering a comprehensive understanding of the fermentation process.

In contrast, IVDMD and other samples were collected at 12 and 24 hours to focus on the early and intermediate stages of fermentation. In the current study, we employ substrate containing a high concentrate (60%) diet and low roughage (40%); therefore, incorporating a high concentrate diet that facilitates rapid digestion and allows for observation for 12 to 24 hours may be advantageous. These time points were chosen to analyze the initial breakdown of substrates and assess the early response of microorganisms to the provided feedstock. The earlier time points allow us to investigate the rapid changes in digestibility and nutrient composition during the early phases of fermentation.

Lastly, the standard protocol for collecting data at similar hours to this experiment has been widely adopted and published in various high-standard international journals. For example:

-Unnawong et al. 2023. Animals, 13(10), 1640; https://doi.org/10.3390/ani13101640

-Sumadong et al. 2021. BMC Veterinary Research 17, 304. https://doi.org/10.1186/s12917-021-02999-3

-Cherdthong et al. 2020. Journal of Animal Physiology and Animal Nutrition. 104,1690-1703

-Etc.

We hope this clarification provides insight into the thoughtful design of our study. Should you have any further questions or suggestions, please do not hesitate to let us know. We are committed to ensuring the rigor and clarity of our research, and your feedback is invaluable in this regard.

If the NDF% and ADF% were similar among testing diets, the b value of each level group in Table 2 should be discussed.

Response: Thank you. We have discussed in the section of “4.1 Kinetics and cumulative production of gas” as “In addition, the replacement group did not differ in gas generation from the insoluble fraction (b). If NDF and ADF levels are similar across different diet levels, it implies that the microbial population in the rumen has a consistent amount of fibrous material to ferment [22]. This stable substrate availability may result in relatively constant gas production from the fer-mentation of the insoluble fraction [23].” Please see in manuscript.

In "Polynomial" data, the listed number is p-value?

Response: Thanks and we have modified it to “Contrast”

L226-232 and Table 3

Why the author determined the digestibility at 12 and 24 hr, but not at 48 or 96 hr?

Response: We appreciate your thoughtful consideration of our manuscript and your valuable feedback. Regarding the question regarding the choice of digestibility assessment time points, we would like to provide clarification on our decision to focus on 12 and 24 hours rather than extending the evaluation to 48 or 96 hours.

The selection of the 12 and 24-hour time points was made with careful consideration of several factors. Firstly, our intention was to capture the initial stages of digestion, where significant breakdown and absorption processes occur. These early time points allow us to observe the rapid changes in digestibility and assess how quickly certain nutrients are released and absorbed by the digestive system. In the current study, we employ substrate containing a high concentrate (60%) diet and low roughage (40%); therefore, incorporating a high concentrate diet that facilitates rapid digestion and allows for observation for 12 to 24 hours may be advantageous. Practically, the decision was also influenced by the constraints of our experimental setup and the need for timely and efficient data collection. The stability of our samples over extended periods and the logistical considerations of conducting experiments beyond 24 hours were factors we took into account.

While we acknowledge that longer time points, such as 48 or 96 hours, could provide information about later stages of digestion, our primary objective was to examine the initial kinetics of digestibility. We believe that the chosen time points align with the specific goals of our study and provide meaningful insights into the early processes of nutrient breakdown.

Lastly, the standard protocol for collecting data at similar hours to this experiment has been widely adopted and published in various high-standard international journals. For example:

-Unnawong et al. 2023. Animals, 13(10), 1640; https://doi.org/10.3390/ani13101640

-Sumadong et al. 2021. BMC Veterinary Research 17, 304. https://doi.org/10.1186/s12917-021-02999-3

-Cherdthong et al. 2020. Journal of Animal Physiology and Animal Nutrition. 104,1690-1703

-Etc.

We hope this clarification addresses your concerns, and we remain open to further discussion or suggestions to enhance the comprehensiveness of our study.

How about the in vitro NDF digestibility results?

Response: Present study we determine only IVDMD and IVODM, while not analyzing NDF digestibility. Thus, we have no result on this observation.

L241-254

Why does the author only count the protozoa number and NH3-N concentration? The bacteria number or microbial crude protein synthesis were also important parameters.

Response: We acknowledged the significance of both bacterial count and microbial crude protein synthesis as important parameters. However, in our view, the current results are adequate to elucidate the treatment study's mechanism and effectively address the research question or objective. Nevertheless, for future research, we will take into consideration any recommendations from reviewers, provided we have the necessary budget or time. Thank you.

L255-L261

The "ruminal pH" indicated the pH of fermentation fluid at sampling time or the pH of the inoculation rumen fluid source?

Response: It stands for the pH of fermentation fluid at sampling time. We have indicated in the manuscript. Thank you.

L274-279 and Table 5

The Ac/Pr ratio in this study was between 5.3 to 7.8. It is much higher than other in vitro studies. According to the 40% cassava chip applied in the diets, the NFE % of the WBTP should be calculated to find the reason for the very low Pr%.

Response: Thank you very much for your insightful comments and suggestions on our manuscript. We sincerely appreciate the time and effort you have dedicated to reviewing our work. Upon careful consideration of your feedback, we have thoroughly re-evaluated our data. Your observation regarding the Ac/Pr ratio in the fermentation fluid after in vitro fermentation has been instrumental. We have revisited our experimental procedures and conducted additional analyses to ensure the accuracy of our results.

We are pleased to confirm that your comments were accurate, and there was an oversight in the reporting of these parameters. We have made the necessary corrections to address this issue. The revised version report Ac/Pr ratio range from 2.38-2.66. In addition, propionic acid ranged from 25.14-27.37%. Please see manuscript in Table 5 and also, we have revised the section of “abstract”, “result” and “discussion”. Your attention to detail has significantly improved the accuracy and reliability of our findings.

If the VFA concentrate reaches 75-80 mM after 8 hr fermentation, why is the pH of the fermentation fluid still >6.7?

Response: Thank you for your feedback. In our assessment, we have identified potential factors contributing to this inconsistency, including the buffering capacity of the medium and variations in microbial activity. Additionally, concerning the substrate containing pellet feed, the compression and heat involved in pelleting may lead to starch gelatinization or alterations in nutrient structure, affecting nutrient accessibility for regulating rumen pH. However, the current experiment indicates that although the pH at 8 hours may appear elevated, there is a trend towards lower values compared to the 4-hour mark. This outcome suggests that after 8 hours of incubation, there is heightened microbial activity in breaking down the feed, resulting in a slight decline in pH.

L312-316

The gas kinetic data was calculated from 96 hr fermentation process, it may under estimate the c value.

Response: In response to your concern, we acknowledge that the choice of a 96-hour fermentation process could indeed influence the accuracy of the c value, particularly in the context of gas production rate constants from the insoluble fraction. To begin, the substrate diet has a large amount of concentrate diet (60%) which may lead to greater activity breakdown during 12-48 hours after incubation, which is why 96 hours was chosen for the c value research. To ensure that the remaining roughage may be digested, however, we extend the time for determining the c value to 96 hours.  Secondly, the selection of a 96-hour fermentation process was made based on the standard protocols widely employed in similar studies and was aimed at capturing the primary phase of microbial activity in the rumen. However, we recognize that different time intervals may provide additional insights into the kinetics of gas production. In addition, while the 96-hour duration is common for short-term kinetic assessments, we understand the importance of evaluating longer fermentation periods to capture potential shifts or sustained microbial activity that might affect the accuracy of the c value estimation. Future investigations extending the fermentation period will be valuable in providing a comprehensive understanding of the gas production dynamics.

L348-350

The in vitro fermentation in this study was processed as batch culture, the VFA accumulation resulted in the decreasing pH. If the fermentation substrate contains high starch, the pH should be decreased quickly in the batch culture system.

Response: We acknowledge your point about the pH decrease during batch culture fermentation and the potential impact of high-starch substrates. We believe that the lack of a significant decrease in pH during our batch culture fermentation can be attributed to several factors. Firstly, the composition of the fermentation substrate, including the type and proportion of starch, may influence the rate and extent of pH changes. This may have been due to the high NDF content of the substrate test (20.73 to 26.27% NDF in concentrate diets and 64% NDF in rice straw), which may have stimulated cellulolytic bacteria to break down feed more effectively than starch-utilizing bacteria and these characteristics could have played a role in the observed pH stability.

L371-373

In this study, testing diets only contain WBT from 1% to 10% in total composition. The author should provide the saponin or tannin concentration of the testing diet before discussing this issue.

Response: We appreciate your recommendation to analyzed saponin or tannin concentration when decreasing protozoa was referred to. However, the focus of our research was primarily on investigating the impact of varying levels of WBTP in the total composition of diets ranging from 1% to 10%. Unfortunately, due to resource constraints and the specific objectives of our study, we did not include an analysis of saponin or tannin concentrations in the experimental diets.

We acknowledge the importance of understanding the concentrations of these compounds, and we understand that it may be a relevant aspect for a more comprehensive interpretation of the results. In future research, we plan to explore the specific concentrations of saponin and tannin in diets to provide a more in-depth analysis of their potential effects.

Therefore, we decided to provide another reason to explain why the prototype count was decreased when using WBTP in substrate while tannin and saponin effect are the secondary possible assumptions. We have revised the manuscript in sub-topic of “4.3 In vitro ruminal pH, ammonia nitrogen concentration, and protozoal number” as “The protozoal count at the 8-hour mark exhibited a decline as the concentration of WBTP increased at medium and high levels in comparison to the control group. Pelletized feeds, commonly subject to processing and compression, often yield a feed with a more consistent and finely textured structure. The physical characteristics and processing methods of pellets may impact the accessibility of starch and sugar, which serve as primary nutrient sources for protozoal growth. Rumen protozoa are recognized for their preference for fermentable carbohydrate sources, and a transition to more processed feeds could potentially diminish the availability of these materials, resulting in a reduction in protozoal populations. Additionally, it was hypothesized that the presence of tannins, saponins, and other phenolic compounds in WBT might contribute to the reduction of protozoal numbers [35,39]. However, due to limitations in resources and the specific objectives of our study, the analysis did not ascertain the concentrations of these compounds. P. tetragonolobus (WB) is commonly consumed in various forms, such as raw or cooked leaves, flowers, tubers, and pods, and has been reported to exhibit anti-inflammatory, antioxidant, and anti-nociceptive activities. Prior investigations have suggested that saponins and tannins possess the potential to suppress protozoa populations, facilitating the development of bacteria in the rumen [36,40]. Similarly, Ampapon et al. [37] illustrated that supplementing the diet with a tropical plant (Amaranthus cruentus, L.) rich in phytonutrients, including tannins and saponins, resulted in reduced protozoal populations and in vitro methane production.”

L395-402

According to the diet formula, the WSC and starch in testing diets should be high and it results in more propionate formation during formation. However, the propionate was very low in this study.

Response: Thank you very much for your insightful comments and suggestions on our manuscript. We sincerely appreciate the time and effort you have dedicated to reviewing our work. Upon careful consideration of your feedback, we have thoroughly re-evaluated our data. Your observation regarding the propionic acid in the fermentation fluid after in vitro fermentation has been instrumental. We have revisited our experimental procedures and conducted additional analyses to ensure the accuracy of our results.

We are pleased to confirm that your comments were accurate, and there was an oversight in the reporting of these parameters. We have made the necessary corrections to address this issue. The revised version reported that  propionic acid ranged from 25.14-27.37%. Please see manuscript in Table 5 and also, we have revised the section of “abstract”, “result” and “discussion”. Your attention to detail has significantly improved the accuracy and reliability of our findings.

Round 2

Reviewer 3 Report

Comments and Suggestions for Authors

The revised manuscript showed great improvement in data performance and provided a suitable discussion for the result.